# West Nile Virus and Usutu Virus: A Post-Mortem Monitoring Study in Wild Birds from Rescue Centers, Central Italy

**DOI:** 10.3390/v14091994

**Published:** 2022-09-09

**Authors:** Giuseppe Giglia, Giulia Mencattelli, Elvio Lepri, Gianfilippo Agliani, Marco Gobbi, Andrea Gröne, Judith M. A. van den Brand, Giovanni Savini, Maria Teresa Mandara

**Affiliations:** 1Division of Pathology, Department of Biomolecular Health Sciences, Faculty of Veterinary Medicine, Utrecht University, 3584 CL Utrecht, The Netherlands; 2Department of Veterinary Medicine, University of Perugia, 06126 Perugia, Italy; 3OIE National Reference Center for West Nile Disease, Istituto Zooprofilattico Sperimentale, dell’Abruzzo e Molise “G. Caporale”, 64100 Teramo, Italy; 4Center Agriculture Food Environment, University of Trento, 38098 San Michele all’Adige, Italy; 5Research and Innovation Centre, Fondazione Edmund Mach, 38098 San Michele all’Adige, Italy; 6Istituto Zooprofilattico Sperimentale dell’Umbria e delle Marche “T. Rosati”, 06126 Perugia, Italy; 7Dutch Wildlife Health Centre, Utrecht University, 3584 CL Utrecht, The Netherlands

**Keywords:** arboviruses, West Nile, usutu, monitoring, wild birds, Central Italy

## Abstract

West Nile virus (WNV) and Usutu virus (USUV) are mosquito-borne flaviviruses that have been associated with neurological diseases in humans and wild birds. Wild bird rescue centers are potential significant hot spots for avian infection surveillance, as recognized in the Italian Integrate National Surveillance Plan for Arboviruses. Here we report the results of a post-mortem active monitoring study conducted from November 2017 to October 2020 on animals hosted in five wild bird rescue centers of Central Italy. Five hundred seventy-six (n = 576) wild birds were tested by real-time polymerase chain reaction (RT-PCR) for the presence of WNV or USUV RNA fragments. No birds tested positive for USUV RNA (n = 0; 0.00%). Evidence of WNV RNA (Ct value = 34.36) was found in one bird (n = 1; 0.17%), an adult little grebe (*Tachybaptus ruficollis* subsp. *ruficollis*), that tested WNV positive in December 2019. This study highlights the strategic role of wildlife rescue centers in monitoring both the introduction and circulation of avian emerging zoonotic diseases. In addition, the presence of WNV during the cold season evidences the possible role of birds in overwintering mechanisms in the Italian territory and requires further investigations.

## 1. Introduction

The family *Flaviviridae* includes several viral species [1]. Most recent outbreaks have been caused by West Nile and Usutu viruses, two species belonging to the *Flavivirus* genus. Included in the Japanese Encephalitis group, they are genetically closely related and share many traits of their transmission cycle which involves birds (amplifying host) and mosquitoes (vector). Mammals, including humans, might be infected through mosquito bites and represent dead-end hosts of these infections. When infected, humans usually develop flu-like syndrome, although sometimes cases of neuroinvasive disease can occur [2,3,4]. In Italy, WNV and USUV infections, sometimes associated with clinical neuroinvasive disease, have been reported in wild birds, horses and humans since the late 1990s [4,5,6,7,8,9,10,11,12,13]. Since 2020, an integrated National Plan for Prevention, Surveillance and Response to Arbovirus (PNA) has been in place as a five-year plan [14] and includes, among other things, the activity of WNV and USUV. PNA aims to early detect viral circulation in Italy to minimize the risk of human infections. In the plan, the Italian territory has been classified in three types of areas (high, low and minimum risk) according to the epidemiological–environmental conditions and, consequently, the transmission risk: (1) High risk areas are those where WNV is circulating or has circulated in at least one of the 5 previous years. (2) Low risk areas are those where WNV has never been or has been rarely reported and where eco-climatic conditions are favorable to viral circulation. (3) Minimum risk areas are those where WNV has never been reported and where eco-climatic conditions are not suitable for WNV circulation [14]. In all three types of areas, the surveillance includes monitoring of wild bird mortality. Wild bird rescue centers (WRCs) are scattered in the territory. They are the main providers of medical support in emergency and wild diseased animal care and may represent an important source of information on WNV and USUV circulation and their virulence characteristics. This study aims to examine carcasses from WRCs of Central Italy to monitor the circulation and introduction of WNV and USUV. Additionally, to obtain possible new insights on the role of wild birds in facilitating the overwintering of USUV and WNV in the study area, bird samples were collected and examined throughout the year.

## 2. Materials and Methods

### 2.1. Animals

Carcasses of wild birds (n = 576) collected by 5 rescue centers located in Umbria, Latium, and Tuscany (Central Italy) from November 2017 to October 2020 were used in this study. All birds died spontaneously or were humanely euthanized for clinical conditions compromising animal welfare (e.g., gunshot lesions, head trauma and fractures). The animals were submitted for necropsy at the Department of Veterinary Medicine, University of Perugia, Umbria, Italy. During necropsy, the cadaver condition was scored as follows: Code 1 (absent autolysis), Code 2 (mild autolysis), Code 3 (moderate autolysis), Code 4 to Code 5 (marked autolysis or corruption, respectively) [15]. For animals with Codes 1 to 3, tissue samples (1 cm^3^) collected during the necropsy were submitted to RT-PCR, histopathology, and immunohistochemistry analyses to investigate either the presence of WNV or USUV or eventual related lesions; for birds with cadaver condition Code 4 or 5, the collected tissues were submitted only for RT-PCR analyses [16]. Ethical approval was not required for this study.

### 2.2. WNV and USUV Molecular Detection

Samples of heart, liver, spleen, kidney, and brain were collected, stored at −80 °C and sent to the WOAH (formerly OIE) Reference Laboratory for West Nile Fever at the Istituto Zooprofilattico Sperimentale “G. Caporale” (Teramo, Italy) for molecular detection of WNV and USUV RNA by RT-PCR analyses. Tissues (brain, heart, liver, spleen, and kidney) were homogenized in phosphate-buffered saline solution. Briefly, after tissue homogenization, RNA extraction was performed by using the MagMAX CORE Nucleic Acid Purification KIT (Applied Biosystem, Termofisher Scientific, Life technologies corporation, Austin, TX, USA), following the manufacturer’s instructions. The extracted RNA was amplified as described in the literature, using a double RT-PCR approach, one aiming for simultaneous detection of WNV lineages 1 and 2 [16] and the other aiming for detection of all WNV lineages [17]. For USUV, the RT-PCR was performed as previously described by Cavrini et al., 2011 [18].

### 2.3. Histopathology and Immunohistochemistry (IHC)

For animals with Code 1 to 3, routine histological examination was performed on formaline-fixed paraffin-embedded (FFPE) tissues collected during necropsy. For the immunohistochemistry, 3 μm sections were obtained from FFPE tissues; the sections were deparaffinized and rehydrated in alcohols. Endogenous peroxidases were blocked by 3% H_2_O_2_ in methanol incubation for 10 min and antigen was retrieved by proteinase K digestion for 10 min at 37 °C. A goat serum block was applied before incubation with the primary antibody. For primary incubation, anti-WNV serum (FLI, Jena, Germany) was used with 1:1700 dilution. As secondary antibody, Bright vision 1 step detection system anti-rabbit HRP was used, with the aminoethyl carbazole (AEC) as substrate. For USUV, IHC was performed as previously described [19]. Primary polyclonal Rabbit Anti-CD3 antibody was used at a dilution of 1:200 (A0452, Dako, Italy) to mark T-cells as a common player of adaptive immunity to intracellular agents in inflammatory infiltrates [20]. As secondary antibody, Mouse Envision HRP kit was used (Ab93697; Abcam, Italy) with the AEC as substrate.

### 2.4. Geographical Distribution Analysis

To map geographical distribution of the rescued animals, collection sites were registered, and coordinates recorded. If the exact collection site was not available, data of the rescue center were used. To visualize the geographical distribution of the collected animals and the location of identified cases, the open-source Quantum Geographic Information System (QGIS^®^) (v. 3.16.10) was used. The geographical distribution of sampling sites and positive case location is shown in Figure 1.

## 3. Results

One out of 576 (0.17%) wild birds tested for the presence of WNV and USUV RNA was found positive for WNV (Ct value: 34.36), while none of the wild birds tested were positive for USUV (Table 1). The only WNV-positive bird was a little grebe (*Tachybaptus ruficollis* subsp. *ruficollis*) belonging to the Order Podicipediformes, Family Podicipedidae. The bird was found on the street side in the Umbria region at the end of December 2019 and admitted to the Veterinary Teaching Hospital of the Department of Veterinary Medicine of the University of Perugia (Italy). It showed depression and difficulties in its movements, partially justified by the presence of a complete luxation of the 3rd phalanx of the 3rd digit, confirmed by radiography. The bird died spontaneously during the day after admission. At necropsy, a mild hepatomegaly and multiorgan congestion were observed. At microscopy, the heart showed mild multifocal loss of cardiomyocytes and interstitial oedema. In the kidneys, tubulo-interstitial nephritis and tubular necrosis were observed, while in the brain, scattered lymphocytes were seen in the meninges and perivascular neuroparenchyma. At immunohistochemistry, leucocyte infiltration consisted of CD3-positive lymphocytes. WNV and USUV antigens were not detected. Additional data regarding the number and percentages of birds collected for the three regions, and for the four seasons of each year of monitoring, are reported in the Appendix A

## 4. Discussion

Epornitic mosquito-borne zoonotic flaviviruses associated with cases in animals and humans have significantly increased their impact on public health in the last decades [21,22]. Their continuous circulation throughout Europe have raised the attention and supported the development of National Surveillance Systems to early detect virus circulation and limit its spread [6]. For mosquito-borne zoonotic flavivirus circulation, birds, being the amplifying hosts, are the major target of surveillance strategies [21,23]. This study reports a post-mortem monitoring on wild birds admitted in WRCs of Central Italy between November 2017 and October 2020. According to the WNV risk classification areas as established in the PNA [14], the WRCs involved in this study were located in the low (Umbria) and in the high-risk areas (Latium and Tuscany). WNV lineage 2 was detected in a little grebe. The little grebe is in the IUCN Red List of Threatened Species (2019) reported as a least-concern species for the risk of extinction. It inhabits small and shallow wetlands, and it is considered resident in Italy. Being a resident species year-round implies that it has probably been infected within the rescue area. In other words, it implies that the infection likely occurred in Umbria. As far as we know, this was the first evidence of WNV circulation in Umbria, an area classified as low risk for WNV circulation. However, looking further, finding a WNV-positive bird in Umbria was somewhat expected, as USUV, which shares the same ecological niche of WNV, has been repeatedly detected by the PNA in this region [24]. Conversely, finding a WNV-positive case at the end of December, when vectors are not flying, was rather surprising. In this case, the histological lesions were mild, antigen could not be detected, and the Ct values were quite high (34.36 cycles). These results, on the one hand, exclude the hypothesis of a recent infection, but on the other hand, they might instead suggest the presence of a persistent infection [25]. Persistent infection has been defined as the detection of a virus in host tissues after viremia has receded [26]. The persistent viral load in organs of birds, and in particular in those belonging to prey species like the little grebe, might sustain the WNV transmission to predators (e.g., birds of prey) months after the mosquito season [27,28]. The little grebe was the only bird out of 576 (0.17%) found positive for WNV RT-qPCR. None of the birds tested positive for USUV RT-qPCR. The lack of WNV and USUV infection cases in high-risk areas (Latium and Tuscany) also during low vector activity periods, additionally support the idea previously suggested in the literature that, contrarily to vectors, birds seem have a minor role in the overwintering of these viruses [29]. Additionally, the high number of negative results and of those found in the high-risk areas might be indicative of the low virulence of most of the WNV and USUV strains circulating in these regions. In the same year of WNV detection in the little grebe, no WNV was detected in mosquitoes as reported from the reports of the OIE national reference lab in Italy (https://www.epicentro.iss.it/westnile/bollettino (accessed on 19 August 2022)). For this reason, our results demonstrate that, even if not ideal to be used alone in a surveillance program because of rescuing only diseased or injured animals, WRCs can be very useful to the National Surveillance Plan in the early detection of the introduction and circulation of emerging zoonotic diseases, improving its sensitivity, expanding the period of surveillance by including the winter months, and providing important information on the virulence of the circulating strains. As a final point, considering that the little grebe is a threatened species, we cannot underestimate that WNV might play a role in this species’ loss.

## 5. Conclusions

For the first time, a case of WNV infection in Umbria (Central Italy), a region currently classified as low-risk area for WNV, was detected during a low-vector activity season (winter). This result suggests that the active monitoring performed through the National Plan of Surveillance on wild birds from wild bird rescue centers can help to better detect the introduction and circulation of WNV. The lack of detection of USUV circulation among mosquitoes and birds needs to be monitored to keep assuring early detection in humans and animals. To ensure WNV circulation and infections of animals and humans are kept under control, additional systematic epidemiological plans of active surveillance on animals received at rescue centers in Italy are highly recommended.

## Figures and Tables

**Figure 1 viruses-14-01994-f001:**
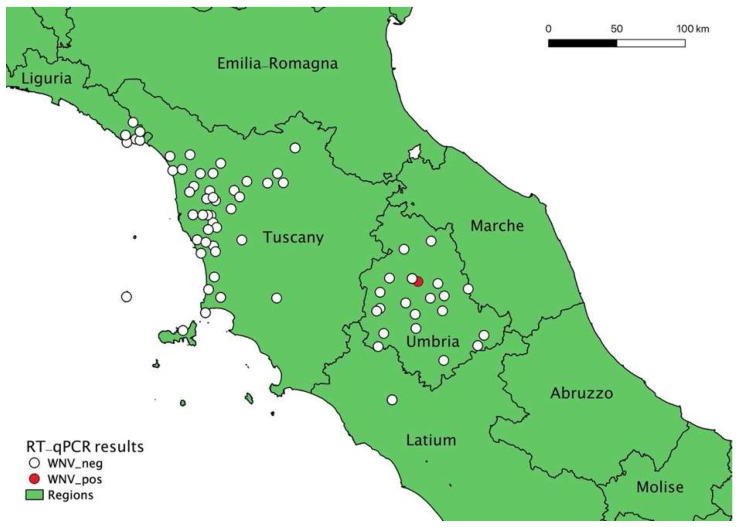
Geographical distribution of sampling sites and positive case location (QGIS map). The map shows the sites of wild birds sampling (white dots) distributed in Umbria, Lazio, Tuscany, and the location of the WNV-positive little grebe (*Tachybaptus ruficollis* subsp. *ruficollis*) (red dot).

**Table 1 viruses-14-01994-t001:** Summary of examined avian species, migratory behavior and WNV RT-PCR results. Data on USUV RT-PCR are not included as results were negative for all the birds.

Order	Family	Species	Migratory/ Resident	WNV RT-PCR
Accipitriformes	Accipitridae	Booted eagle *(Hieraaetus pennatus)*	L	0/1
		Golden eagle *(Aquila chrysaetos)*	R/S	0/2
		Northern goshawk *(Accipiter gentilis)*	R/S	0/3
		Indian vulture *(Gyps indicus)*	R	0/1
		Short-toed snake eagle *(Circaetus gallicus)*	L	0/2
		Western marsh harrier *(Circus aeruginosus)*	P	0/1
		European honey buzzard *(Pernis apivorus)*	L	0/6
		Rough-legged buzzard *(Buteo lagopus)*	S	0/1
		Common buzzard *(Buteo buteo)*	S	0/37
		Harris’s hawk *(Parabuteo unicinctus)*	R	0/1
		Eurasian sparrowhawk *(Accipiter nisus)*	P	0/21
	Pandionidae	Osprey *(Pandion haliaetus)*	R/L	0/1
Anseriformes	Anatidae	Mute swan *(Cygnus olor)*	R	0/1
		Mallard *(Anas platyrhynchos)*	R/S	0/1
Apodiformes	Apodidae	Common swift *(Apus apus)*	L	0/16
Bucerotiformes	Upupidae	Eurasian hoopoe *(Upupa epops)*	R	0/1
Caprimulgiformes	Caprimulgidae	European nightjar *(Caprimulgus europaeus)*	L	0/2
Charadriiformes	Scolopacidae	Woodcocks *(Scolopax rusticola)*	L	0/2
	Laridae	Common gull *(Larus canus)*	R/S	0/28
		Yellow-legged gull *(Larus michahellis)*	R	0/9
Columbiformes	Columbidae	Common wood pigeon *(Columba palumbus)*	P	0/4
		Rock dove *(Columba livia)*	R	0/10
		Eurasian collared dove *(Streptopelia decaocto)*	R	0/7
Coraciiformes	Meropidae	European bee-eater *(Merops apiaster)*	L	0/3
	Alcedinidae	Kingfisher *(Alcedo atthis)*	S	0/1
Falconiformes	Falconidae	Lanner falcon *(Falco biarmicus)*	R	0/1
		Peregrine falcon *(Falco peregrinus)*	R/L	0/8
		Kestrel *(Falco tinnunculus)*	R/L	0/43
		Eurasian hobby *(Falco subbuteo)*	L	0/4
		Merlin *(Falco columbarius)*	R/L	0/1
Gruiformes	Rallidae	Eurasian coot *(Fulica atra)*	R/P	0/1
Passeriformes	Corvidae	Carrion crow *(Corvus corone)*	R	0/25
		Eurasian magpie *(Pica pica)*	R	0/93
		Eurasian jay *(Garrulus glandarius)*	R	0/4
		Western jackdaw *(Coloeus monedula)*	R/P	0/2
	Fringillidae	Canary *(Serinus canaria)*	R	0/12
		European goldfinch *(Carduelis carduelis)*	R/P	0/1
		Hawfinch *(Coccothraustes coccothraustes)*	R/S	0/1
		European greenfinch *(Chloris chloris)*	R	0/1
	Hirundinidae	Common house martin *(Delichon urbicum)*	L	0/5
		Barn swallow *(Hirundo rustica)*	L	0/6
	Muscicapidae	European robin *(Erithacus rubecula)*	R/P	0/8
	Passeridae	House sparrow *(Passer domesticus)*	R	0/17
	Phylloscopidae	Common chiffchaff *(Phylloscopus collybita)*	L	0/1
	Sturnidae	Common starling *(Sturnus vulgaris)*	R/S	0/6
	Sylviidae	Eurasian blackcap *(Sylvia atricapilla)*	P	0/4
	Turdidae	Eurasian blackbird *(Turdus merula)*	R/P	0/16
		Song thrush *(Turdus philomelos)*	P	0/7
Pelecaniformes	Ardeidae	Grey heron *(Ardea cinerea)*	P	0/7
		Cattle egret *(Bubulcus ibis)*	S/L	0/5
		Heron NI	-	0/3
		Eurasian bittern *(Botaurus stellaris)*	P	0/1
Piciformes	Picidae	European green woodpecker *(Picus viridis)*	R	0/7
Podicipediformes	Podicipedidae	Little grebe *(Tachybaptus ruficollis)*	R/S	1/1
Strigiformes	Strigidae	Tawny owl *(Stirx aluco)*	R	0/22
		Eurasian scops owl *(Otus scops)*	L	0/3
		Little owl *(Athene noctua)*	R	0/72
		Long-eared owl *(Asio otus)*	P	0/15
		Eurasian eagle-owl *(Bubo bubo)*	R	0/1
		NI owl	-	0/2
	Tytonidae	Barn owl *(Tyto alba)*	R	0/6
Suliformes	Phalacrocoracidae	Great cormorant *(Phalacrocorax carbo)*	P/L	0/2

NI = Not further identifiable; L = long-distance migrant, S = short-distance migrant, P = partial migrant, Irr = irregular migrant, R = resident.

## Data Availability

Not applicable.

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
