# Peer review of "West Nile Virus and Usutu Virus: A Post-Mortem Monitoring Study in Wild Birds from Rescue Centers, Central Italy"

_viruses, 2022, doi:10.3390/v14091994_

Round 1
Reviewer 1 Report
- The title is misleading - no Usutu virus or its RNA was detected, and only in one bird parts of WNV-2 RNA were detected (but no virus).
- Introduction is too lengthy, and not fully relevant to the topic.
- The only one WNV-RNA-positive bird, Little Grebe (Tachybaptus ruficollis), is not a resident (Table 1, line 163) but a short-migratory species, and it might arrived to Italy from Central Europe in autumn.
- The year of collection of the grebe is missing.
- Many conclusions in the paper are not appropriate, e.g. overwintering of WNV in Umbria, bird-to-bird transmissiomn in Umbria, etc.). In fact, the results of this extensive study do not allow most conclusions drawn (Discussion, Conclusions).
Author Response
Dear reviewer #1,
On behalf of all the authors, I thank you for the time invested looking over the paper and for the comments and suggestion provided to improve the quality of the manuscript.
Below, following each point raised during the peer review (in red), are the responses including the changes made. All changes made in the text are left in “track changes” mode and also highlighted in yellow.
We believe that the manuscript is now improved and suitable for publication in Viruses.
Comments:
- The title is misleading - no Usutu virus or its RNA was detected, and only in one bird parts of WNV-2 RNA were detected (but no virus).
Author’s response: As suggested, changes have been made in the title originally stating: “West Nile virus and Usutu virus in wild birds from Rescue Centers, a post-mortem monitoring study from Central Italy.”. At LINE 2-3 it now states: “West Nile virus and Usutu virus: A post-mortem monitoring study in wild birds from Rescue Centers, Central Italy.”
- Introduction is too lengthy, and not fully relevant to the topic.
Author’s response: As suggested, changes have been made in the introduction. LINE34-43 and 52-56 have been deleted. In addition, small additional changes were made to better improve the readability and give more relevance to the topic of the manuscript.
- The only one WNV-RNA-positive bird, Little Grebe (Tachybaptus ruficollis), is not a resident (Table 1, line 163) but a short-migratory species, and it might arrived to Italy from Central Europe in autumn.
Author’s response: Agreed. The Little Grebe (Tachybaptus ruficollis) can also be considered a short-migratory species based on the area considered (Northern-Eastern Europe). In central Italy (as well as in the rest of the country), the little grebe is considered a resident bird. Data regarding the definition of a resident/short migratory bird were collected from the bird’s page of the IUCN Red List of Threatened Species (https://www.iucnredlist.org/species/22696545/155540155). A change was made in the table to specify also the possible short migrant behavior of this species in some areas.
- The year of collection of the grebe is missing.
Author’s response: the year of collection of the grebe is now added. LINE 197.
- Many conclusions in the paper are not appropriate, e.g. overwintering of WNV in Umbria, bird-to-bird transmissiomn in Umbria, etc.). In fact, the results of this extensive study do not allow most conclusions drawn (Discussion, Conclusions).
Author’s response: None of the following points: overwintering of WNV in Umbria, bird-to-bird transmission in Umbria, are mentioned in the Conclusion paragraph. Conclusions stated are:
(1) For the first time, a case of WNV infection was detected in Umbria (Central Italy).
(2) WNV infection was detected low vector activity season (winter).
(3) The active monitoring performed through the National Plan of Surveillance on wild birds from wild bird rescue centers can help to better detect the introduction and circulation of WNV.
The suggestions mentioned in the revision (overwintering of WNV in Umbria, bird-to-bird transmission in Umbria) were instead raised in the discussion paragraph, all these as possible suggestions based on our data in relation to the literature. For instance, below are additional details on the point raised in the discussion:
- Presence of cases of possible WNV persistent infection described in literature (Ref n 26. Reisen, W.K.; Padgett, K.; Fang, Y.; Woods, L.; Foss, L.; Anderson, J.; Kramer, V. Chronic Infections of West Nile Virus Detected in California Dead Birds. Vector-Borne Zoonotic Dis. 2013, 13, 401–405, doi:10.1089/vbz.2012.1097.)
- Possibility of bird-to-bird transmission (birds of prey/ raptors feeding on infected birds) (ref n29. Vidaña, B., Busquets, N., Napp, S., Pérez-Ramírez, E., Jiménez-Clavero, M. Á., & Johnson, N. (2020). The role of birds of prey in West Nile virus epidemiology. Vaccines, 8(3), 550.)
We do believe that these results if taken into account together, are enough to sustain the points/suggestion raised in the discussion.

Reviewer 2 Report
Giglia and coauthors describe a significant effort to correlate avian mortality with West Nile virus (WNV) and Usutu virus (USUV) infection in Italy. USUV was originally detected in Europe in association with a massive die-off of Blackbirds (Turdus merula), and WNV has increasingly been associated with morbidity in European bird species, such as Magpies (Pica pica) and some raptor species. Despite testing hundreds of carcasses across a wide array of species and four years, only one low-level WNV infection, suggesting a persistent (chronic) infection rather than a fulminating acute infection, was detected in a Little Grebe, a species not considered as an important amplifying host for WNV.
General comments:
11. Background information in the Introduction is disorganized and inappropriate. For example, what do Pestiviruses and hepaciviruses have to do with this research report? Also, Climate change is not directly relevant to the report. Furthermore, I would disagree with the statement “It is common knowledge that climate change is the most important factor influencing the epidemiology and transmission and vector-borne pathogens”. Rather than debating this point, I suggest that you limit the introduction to relevant information needed by the reader to understand the importance of the research you go on to describe.
22. Clarify research objectives and hypothesis. The Introduction should build a case for selecting the research objectives, based on the hypothesis statement. The last sentence of the introduction states that carcasses are tested year-round to investigate the overwintering mechanism of WNV and USUV, yet nothing in the introduction indicated that overwintering was a subject needing investigation.
33. In the Methods, adequate information should be provided for others to repeat the experiment and obtain similar results. Presumably the references cited provide this level of detail. Was there a standardized mass of tissue that was removed or tested from each carcass?
44. Among the results, it is unclear what is the distribution of the seasonality of the observed mortality among the specimens. A simple table showing the sample sizes per month or per quarter would suffice, with carcasses grouped together; or you could add this information to the table to provide this information at the species level.
55. It is not necessary to include the additional column for USUV results in the table, as all are negative. A line of text indicating the negative result would be adequate.
66. In order to interpret these results adequately, it is imperative to know what the surveillance activity and results for WNV and USUV in the mosquito and human/animal populations were in the same regions and periods. If mosquito surveillance was conducted and no virus activity detected, then perhaps the conclusion that avian mortality surveillance was useful for determining the low virulence of WNV/USUV for Italian birds, or possible role of birds in maintaining WNV through winter is appropriate. If alternative surveillance detected virus activity in the region/period of the study, then it would appear that testing bird carcasses for WNV and USUV virus is a major waste of precious resources. I would suggest redirecting those resources toward culturing the avian tissues for viruses (using two or three difference cell culture systems). When cytopathic effects are detected, followup studies can be applied to identify the virus (or other source of pathology) that likely was associated with the mortality.
Specific comments
Line 53. Words associated with the title of the institution should be capitalized.
Line 104. What are “FFPE tissues”?
Line 115. AEC was already introduced on line 111.
Line 162. This sentence does not make sense. You state “For this reason, it is considered resident…” However, habitat preference does not imply residency status. Perhaps you left out the words “throughout the year” or “year-round”
Line 183. “Alternatively” is the wrong word. Consider replacing with “Additionally”
Line 203-205. Extraneous material to be deleted.
Author Response
Dear reviewer #2,
On behalf of all the authors, I thank you for the time invested looking over the paper and for the comments and suggestion provided to improve the quality of the manuscript.
Below, following each point raised during the peer review (in red), are the responses including the changes made. All changes made are also highlighted in yellow in the text.
We believe that the manuscript is now improved and suitable for publication in Viruses.
Giglia and coauthors describe a significant effort to correlate avian mortality with West Nile virus (WNV) and Usutu virus (USUV) infection in Italy. USUV was originally detected in Europe in association with a massive die-off of Blackbirds (Turdus merula), and WNV has increasingly been associated with morbidity in European bird species, such as Magpies (Pica pica) and some raptor species. Despite testing hundreds of carcasses across a wide array of species and four years, only one low-level WNV infection, suggesting a persistent (chronic) infection rather than a fulminating acute infection, was detected in a Little Grebe, a species not considered as an important amplifying host for WNV.
General comments:
- Background information in the Introduction is disorganized and inappropriate. For example, what do Pestiviruses and hepaciviruses have to do with this research report? Also, Climate change is not directly relevant to the report. Furthermore, I would disagree with the statement “It is common knowledge that climate change is the most important factor influencing the epidemiology and transmission and vector-borne pathogens”. Rather than debating this point, I suggest that you limit the introduction to relevant information needed by the reader to understand the importance of the research you go on to describe.
Author’s response: As suggested, changes have been made in the introduction. LINE34-43 and 52-56 have been deleted. In addition, small additional changes were made to better improve the readability and give more relevance to the topic of the manuscript.
- Clarify research objectives and hypothesis. The Introduction should build a case for selecting the research objectives, based on the hypothesis statement. The last sentence of the introduction states that carcasses are tested year-round to investigate the overwintering mechanism of WNV and USUV, yet nothing in the introduction indicated that overwintering was a subject needing investigation.
Author’s response: Changes were made in the text in LINE 100-102 to try to clarify the two objectives of the study,
- Evaluation of wild birds collected in wild animal rescue centers of Central Italy to monitor the circulation and introduction of WNV and USUV in central Italy
- Obtain possible new insights on the role of wild birds in the overwintering of USUV and WNV in the study area
- In the Methods, adequate information should be provided for others to repeat the experiment and obtain similar results. Presumably the references cited provide this level of detail. Was there a standardized mass of tissue that was removed or tested from each carcass?
Author’s response: An addition was made accordingly to the reviewer’s comment to better defined the amount of tissue sampled for each organ, LINE 115.
- Among the results, it is unclear what is the distribution of the seasonality of the observed mortality among the specimens. A simple table showing the sample sizes per month or per quarter would suffice, with carcasses grouped together; or you could add this information to the table to provide this information at the species level.
Author’s response: No outbreaks of avian mortality is reported in the manuscript. The carcasses of birds tested in this monitoring study were actively monitored for WNV and USUV infection, but submitted as part of routinary post-mortem investigation. Clarification on the causes of death of all the 576 birds goes beyond the aim of this manuscript that wants to give focus on the possible role of wild animal rescue centers in the early detect the introduction of new pathogens in a new area and the identification of a case of WNV infection in a region where it has never been reported before, which is currently classified as low risk in the national surveillance plan. As suggested, to add additional information on the percentage of birds examined for each period, details and a clustered bar chart showing the sample sizes per quarter of each year is now added in the supplementary material (Supplementary 1).
- It is not necessary to include the additional column for USUV results in the table, as all are negative. A line of text indicating the negative result would be adequate.
Author’s response: The table 1 (between LINE 214-217) was simplified and reduced, as suggested, eliminating the column reporting the RT-PCR data for USUV, for which all the animals tested were negative. A line of text indicating the negative result for USUV is now added LINE 214.
- In order to interpret these results adequately, it is imperative to know what the surveillance activity and results for WNV and USUV in the mosquito and human/animal populations were in the same regions and periods. If mosquito surveillance was conducted and no virus activity detected, then perhaps the conclusion that avian mortality surveillance was useful for determining the low virulence of WNV/USUV for Italian birds, or possible role of birds in maintaining WNV through winter is appropriate. If alternative surveillance detected virus activity in the region/period of the study, then it would appear that testing bird carcasses for WNV and USUV virus is a major waste of precious resources. I would suggest redirecting those resources toward culturing the avian tissues for viruses (using two or three difference cell culture systems). When cytopathic effects are detected, followup studies can be applied to identify the virus (or other source of pathology) that likely was associated with the mortality.
Author’s response: In the same year, as well as in the year before and after, in area/region where the case of WNV infection was detected in the little grebe (Umbria) no WNV was detected in mosquitoes as reported from the reports of the OIE national reference lab in Italy (https://www.epicentro.iss.it/westnile/bollettino). For this reason, we do believe that the active surveillance on the birds might be of a great help in the early detection of zoonotic emerging infectious diseases affecting wild animals. Data were added in the text in LINE 326-328.
Specific comments
Line 53. Words associated with the title of the institution should be capitalized.
Author’s response: A change was made accordingly to the reviewer’s comment.
Line 104. What are “FFPE tissues”?
Author’s response: A change was made accordingly to the reviewer’s comment. The acronym was specified in the text.
Line 115. AEC was already introduced on line 111.
Author’s response: A change was made accordingly to the reviewer’s comment. AEC was deleted.
Line 162. This sentence does not make sense. You state “For this reason, it is considered resident…” However, habitat preference does not imply residency status. Perhaps you left out the words “throughout the year” or “year-round”
Author’s response: A change was made accordingly to the reviewer’s comment. “year-round” was added in the text.
Line 183. “Alternatively” is the wrong word. Consider replacing with “Additionally”
Author’s response: A change was made accordingly to the reviewer’s comment.
Line 203-205. Extraneous material to be deleted.
Author’s response: At LINE 203-205, the extraneous material was deleted.

Reviewer 3 Report
The manuscript viruses-1876938 „West Nile virus and Usutu virus in wild birds from Rescue Centers, a post-mortem monitoring study from Central Italy” reported the results of one part of a national surveillance plan for Usutu and West Nile virus in Central Italy. Most of the tested dead birds were negative, except one. The manuscript is short and purposeful and therefore in its current form is more suitable for a regional journal. Important information for the scientific interpretation of the results is not included, such as the depiction in which months the examined birds died, increased losses, climate data of the examination period, exact number of examined birds from the three regions. Table 1 is very large and more suitable as supplementary table. However, since only one bird was tested positive, more information about the tested birds and the time point of death should be included.
Minor changings:
Abstract
Page 1, Line 25-26: The binominal name of the little grebe has to be written in italic.
Results
Page 4, Line 131-132: The binominal name of the little grebe has to be written in italic.
Discussion
Page 7, Line 160-161: The binominal name, family and order can be deleted, since it was already written in the result part.
Page 7, Line 166-168: include a reference for this statement
Table 1
Change Kingfishers to Kingfisher
Heron NI not italic
Author Response
Dear reviewer #3,
On behalf of all the authors, I thank you for the time invested looking over the paper and for the comments and suggestion provided to improve the quality of the manuscript.
Below, following each point raised during the peer review (in red), are the responses including the changes made. All changes made are also highlighted in yellow in the text.
We believe that the manuscript is now improved and suitable for publication in Viruses.
Comments:
The manuscript viruses-1876938 „West Nile virus and Usutu virus in wild birds from Rescue Centers, a post-mortem monitoring study from Central Italy” reported the results of one part of a national surveillance plan for Usutu and West Nile virus in Central Italy. Most of the tested dead birds were negative, except one. The manuscript is short and purposeful and therefore in its current form is more suitable for a regional journal. Important information for the scientific interpretation of the results is not included, such as the depiction in which months the examined birds died, increased losses, climate data of the examination period, exact number of examined birds from the three regions.
Author’s response: The main outcome of the paper is the detection of a WNV-infected resident prey bird, in an area classified as low risk from the national surveillance plan, and in a period of low vector activity. The exact number of examined birds from each one of the three regions is now added in the supplementary material (Supplementary 1) in text and pie chart for a better visual representation. Additionally, as suggested from another reviewer (Reviewer #2), to add additional information on the percentage of birds examined for each period, a clustered bar chart showing the sample sizes per quarter of each year is now added in the supplementary material (Supplementary 1).
Table 1 is very large and more suitable as supplementary table.
Author’s response: The table 1 (between LINE 214-217) was simplified and reduced, as suggested from Reviewer #2, eliminating the column reporting the RT-PCR data for USUV, for which all the animals tested were negative. Instead of the column, for USUV a line of text indicating the negative result for the USUV RT-PCR is now added.
However, since only one bird was tested positive, more information about the tested birds and the time point of death should be included.
Author’s response: In our opinion, all the relevant data regarding the clinical signs, post-mortem investigation and molecular and immunohistochemical tests, are included in the manuscript. The time point of death is now added in the text LINE 197.
Minor changings:
Abstract
Page 1, Line 25-26: The binominal name of the little grebe has to be written in italic.
Author’s response: A change was made accordingly to the reviewer’s comment.
Results
Page 4, Line 131-132: The binominal name of the little grebe has to be written in italic.
Author’s response: A change was made accordingly to the reviewer’s comment.
Discussion
Page 7, Line 160-161: The binominal name, family and order can be deleted, since it was already written in the result part.
Author’s response: A change was made accordingly to the reviewer’s comment.
Page 7, Line 166-168: include a reference for this statement
Author’s response: As suggested, a reference for LINE 166-168 is now added at No. 25.
Table 1
Change Kingfishers to Kingfisher
Author’s response: A change was made accordingly to the reviewer’s comment.
Heron NI not italic
Author’s response: A change was made accordingly to the reviewer’s comment.

Round 2
Reviewer 3 Report
All comments and requests were answered sufficiently, thank you!